# Home Cultivation of Cannabis in a Context of Prohibition: Results from Two Online Cross-Sectional Surveys of People Using Cannabis Daily in France

**DOI:** 10.3390/ijerph22081167

**Published:** 2025-07-23

**Authors:** Martin Bastien, Salim Mezaache, Cécile Donadille, Laélia Briand Madrid, Maëla Lebrun, Victor Martin, Perrine Roux

**Affiliations:** 1Sciences Economiques & Sociales de la Santé & Traitement de L’information Médicale—SESSTIM, INSERM, IRD, ISSPAM, Aix-Marseille Université, Faculté des Sciences Médicales et Paramédicales, 13385 Marseille, Franceperrine.roux@inserm.fr (P.R.); 2Association Bus 31/32, 13005 Marseille, France

**Keywords:** cannabis use, home cultivation, cannabis quality, self-regulation, harm reduction, web-based survey, France

## Abstract

In recent decades, European countries have seen a substantial increase in home cultivation of cannabis. In France, the prevalence of cannabis use continues to increase despite its possession, sale, and cultivation being strictly illegal. The present study aimed to describe the profile and motivations of people in France who cultivate cannabis at home. We separately analyzed data from two convenience samples of people who use cannabis daily in France, based on two online cross-sectional surveys. In the first analysis (N = 3840), we used a multivariable logistic regression model to assess factors associated with home cultivation as the main source of cannabis supply. In the second analysis (N = 574), we described participants’ motivations for home cultivation and their cultivation patterns. In the two samples, 11% and 16% reported home cultivation as their main source of supply, respectively. Age, male gender, stable housing, living with a partner, consuming cannabis in herbal form, smoking joints with little or no tobacco, smoking cannabis from a bong or pipe, non-smoking modes of cannabis administration, and using cannabis exclusively for therapeutic reasons were all positively associated with home cultivation, while urban area of residence and at-risk alcohol use were negatively associated. The main reason reported for home cultivation was to manage quality. Few reported selling some of their crop, and most were self-sufficient. Finally, we interpret this practice as a personal response to cannabis prohibition and the unregulated market. Accordingly, possible harm reduction strategies are discussed.

## 1. Introduction

### 1.1. Global Cannabis Market and Home Cultivation

Cannabis is the world’s most widely used illegal drug [1]. Unlike other illegal drugs whose production is localized to specific regions, therefore entailing export on the international market, over the past three decades, cannabis production has extended to every region around the world [1,2]. In Europe, although the cannabis market was dominated until the 2000s by the resin form (hashish) imported from Morocco, the increase in demand for the herbal form led to the development of local production at medium to large-sized sites, mainly organized by criminal groups [2,3]. Data from police seizures show that there is a large illegal cannabis market in Europe and that it is becoming increasingly diverse (e.g., different products, different places of origin, different potencies) [4,5]. In 2020, an estimated 584 tons of resin cannabis and 155 tons of herbal cannabis were seized in European Union member states, an increase of 26% and 19% over 2019 [4,5]. The potency of cannabis products sold globally has also risen over the years, while prices have remained stable; moreover, new types of cannabis products have emerged, such as cannabis concentrates that contain high amounts of Δ-9-tétrahydrocannabinol (THC) (e.g., “wax”, “shatter”) and herbal cannabis products adulterated with synthetic cannabinoids (e.g., “spice”) [4,5].

While the majority of cannabis produced and sold in Europe comes from medium-to-large production sites, the number of people growing small amounts of cannabis independently, with no connection to the illegal market, is increasing [2,3]. In this study, we refer to the practice of small-scale cultivation of cannabis primarily for personal use as ‘home cultivation’. This trend is reflected in a study in the United Kingdom that highlighted that a growing number of police interventions concerned home cultivation involving a small number of plants grown by citizens with no previous criminal record [6]. This increase has been facilitated by the ever-growing availability of information on how to cultivate cannabis and the development of new cultivation technologies and techniques. Several studies have explored the characteristics and motivations of cannabis home cultivators in different countries, including the United Kingdom [7,8], Belgium [9], the Netherlands [9], Denmark [10], Finland [10], Norway [11], Germany [12], Switzerland [12], Austria [12], New Zealand [13], Israel [13], the United States [14], Canada [15], and Italy [16]. The first international online survey examining cannabis cultivation was conducted by the Global Cannabis Cultivation Research Consortium (GCCRC) between 2012 and 2013 in eleven countries [17]. The vast majority of participants were home cultivators, which confirms that this practice is developed throughout Western Europe, North America, and Australia.

With regard to the characteristics of home cultivators, they are generally men, are aged between 22 and 36 years old, are professionally active or are students, and live with their partner or family [17]. In addition, compared with people who use cannabis but who do not grow their own supply, home cultivators tend to be older [14,15], are more likely to have a third-level diploma [15], are less likely to live in urban areas [14], tend to consume cannabis more frequently [14,15], and are more likely to report therapeutic use [15]. With respect to the motivations and attitudes of home cultivators, they grow cannabis primarily for personal use, both to ensure a constant source and to guarantee good quality at an acceptable cost [7,8,9,11,17]. They are generally dissatisfied with cannabis sold on the illegal market, preferring milder, organic products and, in some cases, products that are more suitable for therapeutic use. Most home cultivators are not inclined to increase their production over time, and differ from medium and large-scale growers in terms of financial investment, level of organization, connection with the black market, and cultural values [11]. Furthermore, home cultivators are more ideologically oriented, in the sense that they often embrace the social culture of cannabis, which emphasizes non-violent, ecological, and solidary values and which is characterized by liberal stances on drugs and societal behavior [8,9,11]. Nevertheless, some authors have highlighted a grey zone between ideological and economic motivations for home cultivation: some grow cannabis to save money, and some sell a percentage of their crop for profit through short supply chains [7,8,18].

### 1.2. Drug Policy and Cannabis Use in France

In France, cannabis products containing THC are prohibited. The only exception is medical cannabis, which can be prescribed for a very limited number of conditions, as part of a national experiment that started in 2021 and ended in 2024 [19]. As a result of this experiment, the legal framework to legalize medical cannabis is currently being drawn up. In contrast, low-THC cannabis products, labelled as cannabidiol (CBD) products, are legal and are accessible with little regulation. France has one of the most repressive drug policies for people using cannabis in Europe [20]. French law does not distinguish cannabis from more harmful psychoactive substances in terms of illegality, with a maximum possible sentence for use or possession being a fine of EUR 3750 and one year of imprisonment (Article L.3421-1—*Code de la santé publique*). In practice, however, with a view to simplifying administrative procedures, since 2019, people can be punished by a EUR 200 fixed fine if caught in possession [21]. Moreover, French law penalizes cannabis home cultivation to the same extent as drug trafficking, with a fine of up to EUR 7,500,000 and 20 years of imprisonment (Article 222-35—*Code pénal*), irrespective of the quantity of product produced.

French drug policies have consistently failed to curb the high demand for and supply of cannabis. This is reflected in the fact that the prevalence of cannabis experimentation in the general population is one of the highest in Europe [4] and has increased considerably over the past 30 years [22]. More specifically, in 2023, an estimated 11% of the French general population had used cannabis in the previous 12 months and over 2% were daily users [22]. The strong relationship between demand and supply trends suggests that the importation and local production (i.e., home cultivation or not) of herbal cannabis are constantly increasing and that the herbal form is becoming more common than resin [23]. Approximately two-thirds of people in France who used cannabis in 2020 reported using it in herbal form [24]. In 2005, an estimated 11.5% of the cannabis consumed in France was produced locally [25]. Police data highlight that, between 2010 and 2018, the number of seized plants doubled and that, unlike resin, the percentage of herbal cannabis seizures grew [23]. Data from a 2017 survey of the general population show that the main source of cannabis supply among past-month users was the illegal market (61%), followed by social supply (i.e., a gift or purchase from friends) (32%) [23]. Furthermore, 7% of participants reported home cultivation, although this did not mean it was their exclusive source of supply.

### 1.3. Study Objectives

Despite all the above-mentioned data, there is a lack of data on the characteristics and motivations of home cultivators in France. Given the continuously changing trends in the cannabis market, detailed and recent data on cannabis use patterns are needed to inform harm reduction policy and provide relevant recommendations, specifically in contexts where cannabis use is still prohibited. Accordingly, this present study aimed to identify the sociodemographic profile, drug use patterns, and motivations of people in France who cultivate cannabis at home.

## 2. Materials and Methods

The present study is a secondary analysis of data from two convenience samples of people who use cannabis from two separate surveys, CANNAVID 1 and CANNAVID 2, that were conducted in France between 17 April and 11 May 2020 and between 30 November 2020 and 30 January 2021, respectively, in collaboration with the French harm reduction organization Bus 31/32. The objective of the surveys was to study the impact of lockdown measures implemented in France during the COVID-19 pandemic on the health and drug use patterns in people who use cannabis on a daily basis [26]. Instead, the present study focused specifically on home cultivation practices.

### 2.1. Population and Procedure

CANNAVID 1 and 2 are two national, self-administrated, cross-sectional surveys conducted online in the French language. People living in France (i.e., European French and overseas territories), who were at least 18 years old, and who used cannabis daily before France’s first two COVID-19 lockdowns (i.e., before 17 March 2020 for CANNAVID 1 and before 30 October 2020 for CANNAVID 2) were eligible.

The two survey questionnaires were created on the platform Limesurvey.org. We recruited participants by disseminating each survey through internet forums and websites usually visited by people who use cannabis. After clicking on the survey link, participants received information about its objectives and data protection. The average time for questionnaire completion was 15–20 min. No financial compensation for participation was given. The full CANNAVID 1 questionnaire (in French) is available at the following link: https://doi.org/10.6084/m9.figshare.22183525.v2 (accessed on 15 April 2025). 

To guarantee anonymity, we did not collect any personal data that could identify participants (e.g., names, IP addresses). The medical research ethics committee of the French national institute of health (INSERM) granted ethical approval for both CANNAVID surveys (IRB 00003888, N° 20–676).

### 2.2. Measures

The two survey questionnaires collected the following data:Sociodemographics: Age, gender (man, woman, other), education level (≥upper secondary school, <upper secondary school), professional activity (employed, unemployed, student, retired, disabled), area of residence (based on the number of habitants in the participant’s town of residence: urban (≥100,000), semi-urban (between 100,000 and 10,000), rural (<10,000)), housing type (stable personal, temporary, precarious), having children (yes, no), living with a partner (yes, no), received food aid in the previous month (yes, no).Cannabis use pattern: Main form (herbal, resin, other), main mode of administration (smoked in a joint with a majority of tobacco, smoked in a joint with little or no tobacco, smoked in a bong/pipe, not smoked), daily number of intakes, therapeutic use (non-therapeutic, mixed therapeutic and non-therapeutic, exclusively therapeutic), main source of supply (home cultivation, illegal market, social supply, other).Other psychoactive substance use: At-risk alcohol use (yes, no, based on AUDIT-C score > 3 for men and >2 for women [27]), daily tobacco use (yes, no), past-month opioid use (yes, no), past-month stimulant use (yes, no), past-month benzodiazepine use (yes, no), past-month use of another substance, including ecstasy/MDMA, psychedelics, ketamine, GHB/GBL, and other new psychoactive substances (yes, no)). It should be noted that we did not measure whether these psychoactive substances were prescribed by a medical professional or obtained illegally.

All variables used in the present study refer to the time period before the first two of France’s lockdown periods, specifically before 17 March 2020 for CANNAVID 1 and before 30 October 2020 for CANNAVID 2. It was not mandatory to reply to any question, and participants could answer ‘I do not know’ to each one.

Only age and daily number of cannabis intakes were used as continuous variables. All categorical variables were dichotomized, with the exception of the main mode of cannabis administration and the therapeutic use of cannabis. Moreover, for all the dichotomized categorical variables, we assumed that missing data were not at random; therefore, we assigned ‘missing’ and the ‘I do not know’ responses to one of the categories.

The second CANNAVID survey included more detailed questions than the first on home cultivation patterns for participants who specified ‘home cultivation’ as their main source of supply. We asked them what their main reason was for home cultivation (from a list of the following possible answers based on the community experiences of some of the co-authors: ‘to manage the quality’, ‘to save money’, ‘to make a profit’, ‘to avoid contributing to drug trafficking’, ‘fear of coming into contact with drug dealers’, and an open-text ‘other reason’ option). We also asked them whether they grew cannabis indoors or outdoors, whether they sold some of their crop (yes, no), and whether they were self-sufficient in terms of their own consumption (yes, no).

### 2.3. Statistical Analysis

We analyzed the two datasets (i.e., CANNAVID 1 and CANNAVID 2) separately. They could not be merged as we were not able to identify whether some of the participants had taken part in both surveys. In addition, some of the questions explored in the present study were specific to the CANNAVID 2 survey questionnaire.

CANNAVID 1: From the main supply source variable, we created the outcome for the present study (‘Home cultivation’), which we dichotomized into ‘Yes—home cultivation’ vs. ‘No—other supply sources’. Participants who did not specify their main supply source were excluded from the analysis. We did not take into account missing data for the variables “mode of cannabis administration” and “daily number of intakes”.

We first obtained descriptive statistics of the study sample, stratified on the home cultivation variable. We used the Chi-squared test for categorical variables and the Wilcoxon rank-sum test for non-normally distributed continuous variables to compare the characteristics of participants reporting home cultivation as their main supply source with those who mentioned another source. We then used a multivariable logistic regression model to estimate the adjusted Odds Ratios (aORs) for all explanatory variables associated with home cultivation. To build the multivariable model, all potential explanatory variables associated with home cultivation with a *p*-value threshold < 0.1 in bivariate analyses were eligible to enter the full model. Because of the large number of candidate variables and the resulting risk of overfitting and multicollinearity in the final model, we reduced the number of covariates in the final model. We estimated the final reduced model using a backward selection procedure, removing explanatory variables with the largest *p*-value one at a time until all remaining variables had a *p*-value < 0.05. At each step, we used the Likelihood-Ratio test to ensure that the deletion of each variable did not produce a significant change between estimations of the previous model and the new reduced model. We assessed collinearity in the final model using the Variance Inflation Factor (VIF).

Since people who use the resin form of cannabis are very unlikely to grow cannabis to make their own resin/hash, one can hypothesize that the majority of home cultivators consume herbal cannabis. Therefore, we conducted a sub-analysis by excluding participants who reported using cannabis in resin form and by removing the variable “cannabis form” from the multivariable model.

CANNAVID 2: We described the sociodemographic characteristics and cannabis use patterns of the sample. Among participants who reported home cultivation as their main supply source, we described their main motivation for growing cannabis as well as their cultivation patterns.

All analyses were conducted with STATA-17 (64 bit).

## 3. Results

### 3.1. CANNAVID 1: Factors Associated with Home Cultivation

In CANNAVID 1, of the 4,279 participants who completed the survey, the 3840 living in France who indicated a main source of cannabis supply were included in the present analysis. Eleven percent (10.7%) reported home cultivation as their main source, 60.4% purchased cannabis on the illegal market, 28.1% obtained it socially (i.e., as a gift or purchase from friends), and 0.8% bought it either in shops selling legal low-THC cannabis or in countries neighboring France (e.g., Spanish Cannabis Social Clubs, Dutch Coffee Shops). Participants’ characteristics and substance use patterns according to main supply source (i.e., home cultivation vs. other sources) are given in Table 1.

As shown in Table 1, participants reporting home cultivation as their main supply source mostly identified as men (86.4%), and the median age was 38 (interquartile range (IQR) = 30–45) years old. Most had completed upper secondary education (79.6%) and were professionally active or were studying (79.3%). A large majority (90.8%) had their own stable housing and nearly half (47.7%) lived in an urban area (i.e., in an urban unit with over 100,000 residents). More than half (56.0%) lived with a partner and 45.7% had children. Very few (2.2%) had received food aid during the previous month. Regarding cannabis use patterns, almost all home cultivators (96.1%) reported using cannabis mainly in herbal form, with a median frequency of 4 (IQR = 2–6) intakes per day. The main mode of administration was smoking in joints: 31.2% of participants reporting home cultivation smoked joints with mostly tobacco, while 48.2% smoked joints with little or no tobacco. Other modes of administration were smoking a bong or pipe (3.2%) and vaporization or ingestion (17.4%). Most (64%) home cultivators reported therapeutic use of cannabis; specifically, 21.2% of all those who reported therapeutic use reported exclusive therapeutic use, while 42.8% reported mixed therapeutic and non-therapeutic use. As regards the use of psychoactive substances other than cannabis, half the sample (50.9%) had at-risk alcohol consumption and half (48.4%) were daily tobacco smokers. Ten percent (9.7%) reported using opioids in the previous month, 17.8% reported stimulant use, 5.8% reported benzodiazepine use, and 24.1% reported other substance use.

Table 2 presents the multivariable logistic regression model of factors associated with home cultivation as the main source of cannabis supply. Age (aOR = 1.07 [95%CI = 1.06–1.08]), male gender (aOR = 2.61 [1.86–3.36]), urban area of residence (aOR = 0.36 [0.28–0.46]), personal and stable housing (aOR = 2.17 [1.45–3.24]), living with a partner (aOR = 1.36 [1.06–1.75]), taking cannabis in herbal form (aOR = 17.97 [10.38–31.10), smoking cannabis in joints with little or no tobacco (aOR = 2.63 [2.00–3.47]), smoking a bong or pipe (aOR = 4.10 [1.96–8.58]), modes of administration other than smoking (aOR = 5.67 [3.73–8.61]), exclusive therapeutic use of cannabis (aOR = 1.75 [1.18–2.60]), and at-risk alcohol use (aOR = 0.72 [0.56–0.93]) were all associated with home cultivation. All VIF values were between 1.02 and 1.45, which indicates no problem of collinearity in the final model estimation. In the sub-analysis excluding participants who used cannabis in non-herbal forms (N = 2401), no differences were observed between both multivariable model estimates.

### 3.2. CANNAVID 2: Home Cultivation Patterns

All 574 participants in CANNAVID 2 were included in the present analysis. Table 3 describes their sociodemographic characteristics, their cannabis use patterns, and their cultivation patterns. The main source of cannabis supply was purchasing on the illegal market (59.9%), followed by social supply (16.6%). Sixteen percent reported home cultivation (16.0%), and 1.1% bought cannabis either in shops selling legal low-THC products or in countries neighboring France (e.g., Spanish Cannabis Social Clubs, Dutch Coffee Shops). Six percent (6.1%) did not report a main supply source. Among the 92 participants who cultivated at home, the main motivation reported was to manage the quality of their cannabis (53.3%), followed by avoiding the illegal market (34.8%) (a combination of the ‘to avoid contributing to drug trafficking’ and ‘fear of coming into contact with drug dealers’ modalities (see above)). Economic motivations (a combination of the ‘to save money’ and ‘to make a profit’ modalities (see above)) were reported by 18.5% of participants. As some respondents reported various reasons (i.e., various modalities), they could appear in more than one of these three categories. More than half of home cultivators (55.5%) reported growing cannabis indoors, while 38.0% grew it outdoors. A minority of participants (13.0%) reported selling some of their crop, and a large majority (86.9%) reported that home cultivation left them self-sufficient.

## 4. Discussion

In two separate convenience samples of people living in France who used cannabis daily and who were recruited online in the CANNAVID 1 and 2 surveys, our findings show that a non-negligible proportion (i.e., 11% and 16%, respectively) cultivated it at home. These figures are higher than those in previous studies in the French general population [23]. This is most likely because of our recruitment process, which used specialized websites, and because we restricted inclusion to persons with daily use. Previous studies on the general population showed that the higher the intake frequency, the more likely someone is to grow their own cannabis [23]. Moreover, the internet is a good source to survey hidden populations such as cannabis home cultivators [28].

In our study, gender was related to the likelihood of home cultivation. Specifically, men were more likely than women to cultivate, which reflects findings from a previous European study [29]. Moreover, that same study found that, overall, men and women had different sources for acquiring psychoactive substances (i.e., cannabis and other drugs). In addition, several authors have described gender-specific cultural norms that limit not only cannabis consumption among women but also their ability to participate in supply dynamics (buying, selling, and growing) and their inclusion in cannabis user networks [30,31,32].

In our study, participants who cultivated at home were more likely to be older, to live with their partner, to have stable personal housing, and to live in a non-urban area. Non-urban areas and stable personal housing are favorable conditions for home cultivation, since people have the space to grow and are less likely to be discovered by law enforcers. Furthermore, taken together, being older, living with a partner, and having stable personal housing could suggest a more socially integrated way of life. These results reflect those from a previous study that showed that many cannabis cultivators “*are people who live more-or-less normal lives rather than some deviant or anti-social sub-group*” [17] (p. 235). Compared with other participants, those who cultivated at home were also more likely to report less risky modes of consumption (i.e., smoking less tobacco in joints and using administration routes other than smoking) and were less likely to report at-risk alcohol use. These lower risk patterns of substance use may be related to the fact that a substantial proportion of our sample were older and that a substantial proportion reported exclusively therapeutic cannabis use compared with participants who did not cultivate at home. Moreover, home cultivators were more likely to smoke from a bong or pipe, although the number of participants who did so was very low. This may be linked to a desire to consume a greater quantity at each intake; however, it cannot be considered a safer pattern of use. A previous study showed that many home cultivators reported that they grew cannabis for self-medication for a wide variety of conditions [33]. These persons tend to be even more concerned about the composition and the cannabinoid concentration of the cannabis they consume compared with other home cultivators [34]. In terms of more responsible patterns of cannabis use in older cultivators, previous studies revealed that, as persons get older, they change both their consumption pattern and how they manage the effects of cannabis according to daily life responsibilities [35,36]. Many people who use drugs function normally in their daily life thanks to awareness of possible associated harms and adopting strategies to prevent them [37]. Cultivating cannabis at home may be one such strategy.

Interestingly, the main reported motivation for home cultivation in our study was to manage the quality of cannabis. People who use cannabis usually assess cannabis quality according to sensory criteria, the effects they experience, and perceived safety, all of which depend on composition (e.g., cannabinoid content, absence of adulterants) and the context of production (e.g., geographic origin, cultivation techniques). For example, users usually perceive locally produced organic cannabis to be of satisfactory quality [38]. Cannabis quality cannot be certified in an unregulated market [39]. By cultivating cannabis at home, growers can control the composition and the context of production to a certain extent. The second most frequently reported motivation for home cultivation was avoiding the illegal market, which exposes cannabis users to a greater risk of legal repercussions as well as stigma associated with drug trafficking. In comparison, economic motivations (i.e., saving money and making a profit from sales) were much less frequently reported by participants in the study who cultivated at home. This would suggest that home cultivation is motivated more by safety and moral reasons than by economic ones. The fact that home cultivators generally did not sell cannabis and that crops were sufficient to ensure at least their personal consumption supports this assumption. Our results may reflect the description of “ideologically-oriented cultivators” (as opposed to “commercially-oriented cultivators”) found in the literature; these persons grow a small crop primarily for their own consumption and for their close social circle, are concerned about the quality of cannabis, and distance themselves from the illegal cannabis market and commercial practices [8,9,17]. Our results suggest that home cultivation of cannabis can be understood as a conscious strategy to reduce some of the negative consequences of cannabis prohibition, such as unknown quality, an illegal market led by criminal groups, and criminalization of users [9,12,17]. In addition, previous studies reported that home cultivators tend to be relatively law abiding outside of the fact that they illegally grow and use cannabis [12,17]. On the basis of these results, understanding this practice solely as a criminal activity would fail to consider its possible benefits and the motivations of home cultivators. We consider that the criminalization of home cultivation of cannabis for personal use and associated legal sanctions (e.g., in France, up to EUR 7,500,000 in fines and 20 years in prison) are inappropriate and disproportionate, precisely because of the fact that many home cultivators seek to distance themselves from the illegal market and its harmful aspects. Based on similar results, some authors have recommended that the criminal justice systems make a distinction between persons who cultivate for personal use and commercial cultivators, based on the number of plants they grow and whether they are involved in drug trafficking or not [7,9,12].

Finally, regulatory strategies could potentially reduce the harms associated with the illegal cannabis market [40,41]. Cannabis policy choices have an impact on cannabis quality and prices, as well as on how people obtain their products [42,43]. Strategies based on self-regulation and on market regulation are two possible avenues for safer cannabis supply in France.

First, self-regulation strategies could foster spontaneous behaviors of cannabis users who value the quality of cannabis products in terms of health self-management [40,41,44]. For example, some countries—like Spain and Belgium—have decriminalized (de facto or de jure) home cultivation for personal use with a view to interrupting cannabis supply from illegal drug markets. Other countries—such as Uruguay and Canada—have legalized home cultivation as one supply option. However, decriminalization and legalization models vary greatly in terms of permitted cultivation (the limit on the number of plants one can grow, places of cultivation, the possibility of sharing one’s produce on a not-for-profit basis, etc.) [44]. In jurisdictions where home cultivation is authorized, people who use cannabis are more likely to grow their own crops [45,46]. The legal consequences of cannabis cultivation in many countries, especially the risk of being discovered and prosecuted, are the main concern for people who grow cannabis at home [12] and are probably an important factor in the decision by many cannabis users not to cultivate. Besides home cultivation for personal use, the Cannabis Social Club (CSC) model provides another example of self-regulation of cannabis consumption at the community level, through mutual aid and a set of formal and informal rules [47]. CSCs are associations of people that use cannabis who collectively organize cultivation and who provide harm reduction education to their members [48]. Different CSC models have been described in various countries, including Spain, Belgium, and Uruguay [49].

Second, market regulation strategies could focus on impacting supply dynamics and improving the safety of cannabis products [40,41]. For example, drug-checking services help to monitor the quality of illicit drugs at various geographic levels (e.g., local, regional, national) and act as a warning system for poor-quality products in circulation [50]. If users are provided with information about drug composition, they can adapt their consumption behaviors and choose safer sources [50]. With regard to cannabis products, drug-checking services could highlight the presence of synthetic cannabinoids [51]. Home cultivators could also benefit from these services since they are rarely able to accurately assess cannabinoid content themselves. However, few drug-checking services adapted to cannabis products are currently available for consumers in France. Moreover, in the context of legalized cannabis markets, regulatory authorities could enhance the safety of legally available products by imposing production standards and quality controls on licensed producers [52,53]. More specifically, cannabinoid content and the presence of toxic contaminants (pesticides, solvents, heavy metals, bacteria, etc.) could be regulated with a view to protecting consumer health. In addition, packaging could communicate reliable information on composition and potency and be transparent about the conditions of production [54].

### Limitations

This study has limitations. First, because the recruitment process used specialized websites, and because we restricted inclusion to persons with daily use, our results cannot be generalized to all persons who cultivate cannabis at home in France. It is possible that some home cultivators may have limited access to the internet. It is also possible that non-daily users of cannabis (e.g., occasional users) grow cannabis at home. In addition, more than 6% (CANNAVID 1) and 8% (CANNAVID 2) of participants did not answer the ‘main source of supply’ question, suggesting that, for some users, this is sensitive information. Accordingly, our results may underestimate the number of home cultivators in both surveys. Furthermore, we cannot exclude the possibility of misclassification of participants who grew their own cannabis but not enough for it to be considered their main source of supply.

Second, this study provides only a limited understanding of home cultivation practices. Because the study design was cross-sectional, we were not able to identify the direction of the association between participants’ characteristics and the reported main supply source. In addition, since describing home cultivation practices was not the initial objective of the two CANNAVID surveys, only a very small number of questions about cultivation patterns and associated perceptions were present in both surveys’ questionnaires. Further studies could provide a deeper understanding of both the temporal development of home cultivation in the individual consumption trajectory and the perceptions of benefits and risks associated with this practice.

## 5. Conclusions

In the global context of the increasing use and accessibility of cannabis, home cultivation is becoming increasingly common among people who use cannabis. This study of daily cannabis users in France, which was conducted in collaboration with a community health association, provides data on the profiles and motivations of cannabis home cultivators. We found that a non-negligible proportion of people using cannabis in France obtain cannabis by growing it at home, without relying on or contributing to the illegal market. Compared with other cannabis users, these home cultivators tend to have more stable and socially integrated lifestyles as well as more responsible substance use patterns. In line with previous similar studies, we interpret this practice as a personal strategy to reduce some of the harms related to cannabis prohibition and the illegal market. Specifically, home cultivators seem to be primarily concerned with product quality and less concerned with economic or commercial considerations. Consequently, considering this practice to be solely a criminal activity fails to understand its possible benefits and the motivations of home cultivators. Since the possession, sale, and cultivation of cannabis remain strictly illegal in France, this study questions the relevance of drug policies that criminalize the cultivation of cannabis for personal use. Instead, this study supports the implementation of harm reduction policies that foster self-regulatory behaviors among people who use cannabis.

## Figures and Tables

**Table 1 ijerph-22-01167-t001:** Characteristics and substances use patterns of participants (N = 3840) from the CANNAVID 1 survey, stratified by home cultivation (Yes, N = 411; No, N = 3429), Chi-squared test or Wilcoxon rank-sum test.

Variable	Total N (%) Median [IQ]	Home Cultivation		*p*-Value
		Yes	No	
Age (continuous)	27 [22,23,24,25,26,27,28,29,30,31,32,33,34,35,36]	38 [30,31,32,33,34,35,36,37,38,39,40,41,42,43,44,45]	26 [22,23,24,25,26,27,28,29,30,31,32,33,34]	<0.001
Gender Women and other Men	1028 (26.8) 2812 (73.2)	56 (13.6) 355 (86.4)	972 (28.3) 2457 (71.7)	<0.001
Type of area of residence Rural, semi-urban, and unknown * Urban	1325 (34.5) 2515 (65.5)	215 (52.3) 196 (47.7)	1110 (32.4) 2319 (67.6)	<0.001
Education level <Upper secondary school and unknown* ≥Upper secondary school	666 (17.3) 3174 (82.7)	84 (20.4) 327 (79.6)	582 (17.0) 2847 (83.0)	0.080
Housing type ^1^ Temporary, precarious, and unknown * Personal stable housing	948 (24.7) 2892 (75.3)	38 (9.2) 373 (90.8)	910 (26.5) 2519 (73.5)	<0.001
Living with a partner No and unknown * Yes	2197 (57.2) 1643 (42.8)	181 (44.0) 230 (56.0)	2016 (58.8) 1413 (41.2)	<0.001
Having children No and unknown * Yes	2940 (76.6) 900 (23.4)	223 (54.3) 188 (45.7)	2717 (79.2) 712 (20.8)	<0.001
Professional activity Unemployed, retired, disabled, and unknown * Employed or student	606 (15.8) 3234 (84.2)	85 (20.7) 326 (79.3)	521 (15.2) 2908 (84.8)	0.004
Received food aid in the past month ^1^ No and unknown * Yes	3749 (97.6) 91 (2.4)	402 (97.8) 9 (2.2)	3347 (97.6) 82 (2.4)	0.800
Cannabis form ^1^ Resin, other (oil, concentrates), and unknown * Herbal	1435 (37.4) 2405 (62.6)	16 (3.9) 395 (96.1)	1419 (41.4) 2010 (58.6)	<0.001
Mode of administration ^1,2^ Smoked in a joint w/mostly tobacco Smoked in a joint w/little or no tobacco Smoked in a bong or pipe Not smoked (vaporization, ingestion)	2354 (61.4) 1230 (32.1) 91 (2.4) 159 (4.1)	127 (31.2) 196 (48.2) 13 (3.2) 71 (17.4)	2227 (65.0) 1034 (30.2) 78 (2.3) 88 (2.6)	<0.001
Daily number of intakes ^1,3^ (continuous)	4 [3,4,5,6]	4 [2,3,4,5,6]	4 [3,4,5,6]	0.929
Therapeutic use ^1^ Non-therapeutic Mixed therapeutic and non-therapeutic Exclusively therapeutic Unknown *	1156 (30.1) 1521 (39.6) 407 (10.6) 756 (19.7)	91 (22.1) 176 (42.8) 87 (21.2) 57 (13.9)	1065 (31.1) 1345 (39.2) 320 (9.3) 699 (20.4)	<0.001
At-risk alcohol use ^1,4^ No and unknown * Yes	1388 (36.1) 2452 (63.9)	202 (49.1) 209 (50.9)	1186 (34.6) 2243 (65.4)	<0.001
Daily tobacco use ^1^ No and unknown * Yes	1539 (40.1) 2301 (59.9)	212 (51.6) 199 (48.4)	1327 (38.7) 2102 (61.3)	<0.001
Opioid use in the past month ^1,5,6^ No and unknown * Yes	3562 (92.8) 278 (7.2)	371 (90.3) 40 (9.7)	3191 (93.1) 238 (6.9)	0.039
Stimulant use in the past month ^1,5,7^ No and unknown * Yes	3071 (80.0) 769 (20.0)	338 (82.2) 73 (17.8)	2733 (79.7) 696 (20.3)	0.225
Benzodiazepine use in the past month ^1,5^ No and unknown* Yes	3590 (93.5) 250 (6.5)	387 (94.2) 24 (5.8)	3203 (93.4) 226 (6.6)	0.560
Other substance use in the past month ^1,5,8^ No and unknown * Yes	2748 (71.6) 1092 (28.4)	312 (75.9) 99 (24.1)	2436 (71.0) 993 (29.0)	0.039

* Unknown refers to missing and ‘I do not know’ responses. ^1^ Refers to the month before the COVID-19-related lockdown in France (17 March 2020). ^2^ N = 3834. ^3^ N = 3801. ^4^ AUDIT-C score > 3 for men and >2 for women. ^5^ Prescribed or not. ^6^ Opioids include heroin, analgesic opioids (e.g., codeine, tramadol, morphine), and opioid substitution treatments (e.g., methadone, buprenorphine). ^7^ Stimulants include cocaine hydrochloride, crack, amphetamine, and methylphenidate. ^8^ Other substances include ecstasy/MDMA, ketamine, psychedelics, GHB/GBL, and new psychoactive substances.

**Table 2 ijerph-22-01167-t002:** Factors associated with home cultivation in participants from the CANNAVID 1 survey, multivariable logistic regression (reduced model), N = 3834.

Variable	Adjusted Odds Ratio (aOR)	95% Confidence Interval (95%CI)	*p*-Value
Age (per year increase)	1.07	[1.06–1.08]	<0.001
Gender: men (Ref: women and other)	2.61	[1.86–3.66]	<0.001
Area of residence: urban (Ref: rural, semi-urban, and unknown) *	0.36	[0.28–0.46]	<0.001
Housing type: stable personal (Ref: temporary, precarious, and unknown) *	2.17	[1.45–3.24]	<0.001
Living with a partner: yes (Ref: no and unknown) *	1.36	[1.06–1.75]	0.015
Cannabis form ^1^: herbal (Ref: resin, other (oil, concentrates), and unknown) *	17.97	[10.38–31.10]	<0.001
Mode of administration ^1^: Smoked in a joint w/little or no tobacco Smoked in a bong or pipe Not smoked (vaporization, ingestion) (Ref: smoked in joint w/mostly tobacco)	2.63 4.10 5.67	[2.00–3.47] [1.96–8.58] [3.73–8.61]	<0.001 <0.001 <0.001
Therapeutic use ^1^: Mixed therapeutic and non-therapeutic Exclusively therapeutic Unknown * (Ref: non-therapeutic)	1.25 1.75 0.98	[0.92–1.71] [1.18–2.60] [0.66–1.45]	0.160 0.006 0.908
At-risk alcohol use ^1,2^: yes (Ref: no and unknown) *	0.72	[0.56–0.93]	0.010

* Unknown refers to missing and ‘I do not know’ responses. ^1^ Refers to the month before the COVID-19-related lockdown in France (17 March 2020). ^2^ AUDIT-C score >3 for men and >2 for women.

**Table 3 ijerph-22-01167-t003:** Characteristics and cannabis use patterns of participants from the CANNAVID 2 survey, N = 574.

Variable	Total N (%)
Age (med, IQR) (years)	26 [22,23,24,25,26,27,28,29,30,31,32,33,34,35,36]
Gender: men	438 (76.3)
Urban area (>100,000 residents)	380 (66.2)
Education level ≥ upper secondary school	474 (82.6)
Stable personal housing	428 (74.6)
Living with a partner	232 (40.4)
Having children	123 (21.4)
Employed or student	423 (73.7)
Food aid in the past month	12 (2.1)
Cannabis form: herbal	404 (70.4)
Mode of administration Smoked in a joint w/mostly tobacco Smoked in a joint w/little or no tobacco Smoked in a bong or pipe Not smoked (vaporization, ingestion)	381 (66.4) 127 (22.1) 20 (3.5) 40 (7.0)
Daily number of intakes (med, IQR)	4 [2,3,4,5]
Therapeutic use Non therapeutic Mixed therapeutic and non-therapeutic Exclusively therapeutic Unknown *	131 (22.8) 333 (58.0) 63 (11.0) 47 (8.2)
Source of cannabis supply Home cultivation Illegal market Social supply Other countries and legal supply Unknown *	92 (16.0) 344 (59.9) 97 (16.9) 6 (1.1) 35 (6.1)
Home cultivation: Main reason (N = 92) To manage quality To avoid the illegal market To save money or to make profit Other and unknown *	49 (53.3) 32 (34.8) 17 (18.5) 2 (2.2)
Home cultivation: Growing method (N = 92) Indoors Outdoors Unknown *	51 (55.5) 35 (38.0) 6 (6.5)
Home cultivation: Selling some of their crop (N = 92) No Yes Unknown *	78 (84.8) 12 (13.0) 2 (2.2)
Home cultivation: Self-sufficient (N = 92) No Yes Unknown *	10 (10.9) 80 (86.9) 2 (2.2)

* Unknown refers to missing and ‘I do not know’ responses.

## Data Availability

The data presented in this study are available upon request from the authors. Data are not publicly available due to privacy and ethical restrictions.

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
