# Peer review of "Home Cultivation of Cannabis in a Context of Prohibition: Results from Two Online Cross-Sectional Surveys of People Using Cannabis Daily in France"

_ijerph, 2025, doi:10.3390/ijerph22081167_

Round 1

Reviewer 1 Report

Comments and Suggestions for Authors

Review for International Journal of Environmental Research and Public Health

Home cultivation of cannabis in a context of prohibition: results from two online cross-sectional surveys of people using cannabis daily in France.

In the reviewed article, drawing on two survey datasets, the authors examine the correlates of home cannabis cultivation in France. My opinion of the article is positive. I have some suggestions to strengthen the article. The biggest issue I would like the authors to address is to introduce the correlates (and their theoretical logic), even if briefly, in the front-end. Otherwise, the methods and findings can be overwhelming.

Front-End

  • The front-end is well written, clear, and appears to cover the relevant research. However, I wonder if the authors might develop an additional section where they provide some a priori expectations for which correlates they expect to be associated with home cultivation. I am NOT asking for hypotheses, but some theoretical expectations that will gently introduce the readers to the correlates and a general logic for the associations – even if it’s just for some of the correlates; otherwise, the methods and findings can be overwhelming when all the correlates are first introduced there.

  • Note: the authors do this in the discussion. So, again, just a paragraph on the front-end that offers a brief introduction to these correlates and their logic. Then the discussion could be used to more fully flesh out these theoretical connections.

Methods

  • Is CANNAVID an acronym? It might be good to state the full name at least once.
  • The authors should state that this was a convenience sample. I know it’s fairly obvious from the description, but it should be explicitly stated.
  • Is there a concern of a selection effect with this sampling approach, especially given where the authors drew their sample? Are the cannabis users in these online venues representative of cannabis users broadly? If so, it’s not a significant drawback, but the authors might consider mentioning it in the discussion as a limitation. Note: the authors address this in the discussion.
  • In the measures section, many of the other psychoactive substances are prescribed substances. Are the authors measuring use generally or misuse in particular? The wording suggests use generally (which is fine), but I would like the authors to confirm.
  • Line 162: should “situation” be replaced with “time period”?
  • Why the backward selection procedure when building the model for the logistic regression analysis of the CANNAVID 1 data? It seems there was sufficient statistical power to include all the covariates. Please elaborate on why you chose this modeling method.

Findings

  • This is a lot to absorb. I reemphasize the importance of introducing at least some of these correlates in the front end. Otherwise, this is a reasonably clear presentation of the findings.
  • Unknown is misspelled in each of the tables’ notes.
  • In Table 1, continuous is misspelled as continue.

Discussion

  • It seems odd to start out the discussion by talking about the study’s methodological limitations. I would suggest moving that chunk to before the conclusion. However, if this is a disciplinary difference between the authors and myself, I don’t mind if the authors keep it as it is.
  • The discussion effectively ties the findings to previous work.
  • In the first paragraph addressing harm reduction and public policy, the authors shift the format of their in-text citations to APA. Otherwise, this section is also good. Good policy suggestions.

Author Response

Comment 1 : The front-end is well written, clear, and appears to cover the relevant research. However, I wonder if the authors might develop an additional section where they provide some a priori expectations for which correlates they expect to be associated with home cultivation. I am NOT asking for hypotheses, but some theoretical expectations that will gently introduce the readers to the correlates and a general logic for the associations – even if it’s just for some of the correlates; otherwise, the methods and findings can be overwhelming when all the correlates are first introduced there.

Note: the authors do this in the discussion. So, again, just a paragraph on the front-end that offers a brief introduction to these correlates and their logic. Then the discussion could be used to more fully flesh out these theoretical connections.

Response 1 : Thank you for your comment. As per your suggestion, we have added the following sentences to the introduction section (page 2, lines 70-76): “With regard to the characteristics of home cultivators, they are generally men, are aged between 22 and 36 years old, are professionally active or are students, and live with their partner or family [17]. In addition, compared to people who use cannabis but who do not grow their own supply, home cultivators tend to be older [14,15], are more likely to have a third level diploma [15], are less likely to live in urban areas [14], tend to consume cannabis more frequently [14,15], and are more likely to report therapeutic use [15]. With respect to the motivations and attitudes of home cultivators, they grow cannabis primarily for personal use […]”.

Comment 2 : Is CANNAVID an acronym? It might be good to state the full name at least once.

Response 2 : CANNAVID is not an acronym.

Comment 3 : The authors should state that this was a convenience sample. I know it’s fairly obvious from the description, but it should be explicitly stated.

Response 3 : Thank you for this suggestion. We added this detail in three sections, as follows:

-Abstract (page 1, line 18): “We separately analyzed data from two convenience samples of people who use cannabis daily in France, based on two online cross-sectional surveys.”

-Methods (page 3, lines 132-133): “The present study is a secondary analysis of data from two convenience samples of people who use cannabis from two separate surveys, CANNAVID 1 and CANNAVID 2, …”.

-Discussion (page 10, line 309): “In two separate convenience samples of people living in France who used cannabis daily and who were recruited online in the CANNAVID 1 and 2 surveys, our findings […]”.

Comment 4 : Is there a concern of a selection effect with this sampling approach, especially given where the authors drew their sample? Are the cannabis users in these online venues representative of cannabis users broadly? If so, it’s not a significant drawback, but the authors might consider mentioning it in the discussion as a limitation.

Note: the authors address this in the discussion.

Response 4 : As this issue is addressed as a limitation in the discussion section, we do not believe it is necessary to mention it in the Methods section.

Comment 5 : In the measures section, many of the other psychoactive substances are prescribed substances. Are the authors measuring use generally or misuse in particular? The wording suggests use generally (which is fine), but I would like the authors to confirm.

Response 5 : We did not measure whether the psychoactive substances mentioned were prescribed or not, so it is ‘general use’. To clarify this, we added the following sentence in the measures section (page 4, lines 174-176):It should be noted that we did not measure whether these psychoactive substances were prescribed by a medical professional or obtained illegally”.

We also added the words “Prescribed or not” in Table 1’s legend for all substances that can be prescribed (page 7 line 247).

Comment 6 : Line 162: should “situation” be replaced with “time period”?

Response 6 : Thank you for this suggestion; we replaced “situation” with “time period” (page 4, line 177).

Comment 7 : Why the backward selection procedure when building the model for the logistic regression analysis of the CANNAVID 1 data? It seems there was sufficient statistical power to include all the covariates. Please elaborate on why you chose this modeling method.

Response 7 : In our multivariable regression modeling, thanks to the large sample size, we had enough statistical power to retain many candidate covariates in the final model. Despite this, we chose to use the backward selection procedure in order to obtain a final reduced model that would be simpler, more parsimonious, and have a smaller number of covariates, with a view to reducing the risk of multicollinearity between covariates or overfitting of the model. We added this justification in the method section (page 5, lines 214-216) as follows: “[…] all potential explanatory variables associated with home cultivation with a p-value threshold <0.1 in bivariate analyses were eligible to enter the full model. Because of the large number of candidate variables and the resulting risk of overfitting and multicollinearity in the model, we reduced the number of covariates in the final model. We estimated the final reduced model using a backward selection procedure […]”.

Comment 8 : Findings: This is a lot to absorb. I reemphasize the importance of introducing at least some of these correlates in the front end. Otherwise, this is a reasonably clear presentation of the findings.

Response 8 : We introduced correlates in the introduction section, as suggested (page 2, lines 70-76).

Comment 9 : Unknown is misspelled in each of the tables’ notes.

Response 9 : Thank you for pointing out this error. This has been corrected on page 7 line 245, page 8 line 285, and page 9 line 307.

Comment 10 : In Table 1, continuous is misspelled as continue.

Response 10 : Thank you for pointing out this error. This has been corrected in two places in Table 1 on page 6.

Comment 11 : It seems odd to start out the discussion by talking about the study’s methodological limitations. I would suggest moving that chunk to before the conclusion. However, if this is a disciplinary difference between the authors and myself, I don’t mind if the authors keep it as it is.

Response 11 : Thank you for this comment. This was a personal choice of the first author, but we agree with your suggestion and have moved the limitations to a separate sub-section at the end of the discussion section, as follows (page 12 lines 430-450): “Limitations. This study has limitations. First, because the recruitment process used specialized websites, and because we restricted inclusion to persons with daily use, our results cannot be generalized to all persons who cultivate cannabis at home in France. It is possible that some home cultivators may have limited access to the internet. It is also possible that non-daily users of cannabis (e.g., occasional users) grow cannabis at home. In addition, more than 6% (CANNAVID 1) and 8% (CANNAVID 2) of participants did not answer the ‘main source of supply’ question, suggesting that for some users, this is sensitive information. Accordingly, our results may underestimate the number of home cultivators in both surveys. Furthermore, we cannot exclude the possibility of misclassification of participants who grew their own cannabis but not enough for it to be considered their main source of supply.

Second, this study provides only a limited understanding of home cultivation practices. Because the study design was cross-sectional, we were not able to identify the direction of the association between participants’ characteristics and the reported main supply source. In addition, since describing home cultivation practices was not the initial objective of the two CANNAVID surveys, only a very small number of questions about cultivation patterns and associated perceptions were present in both surveys’ questionnaires. Further studies could provide a deeper understanding of both the temporal development of home cultivation in the individual consumption trajectory, and the perceptions of benefits and risks associated with this practice.”.

Comment 12 : In the first paragraph addressing harm reduction and public policy, the authors shift the format of their in-text citations to APA. Otherwise, this section is also good. Good policy suggestions.

Response 12 : Thank you for this comment and for pointing out this error. We corrected the format of the citations (page 11, lines 378-379).

Reviewer 2 Report

Comments and Suggestions for Authors

I have enjoyed reading this article on home cultivation of cannabis and have only one minor issue and a few typos to point to.

  1. 11 line 375-6. “home cultivation for personal use should not be considered as a criminal activity”. As much as I agree with this statement, it seems odd to present it basically out of nowhere. Why should it not be thus considered? Please explain or soften the categorical policy recommendation somewhat (e.g., “In context of the findings from these surveys, the authors believe that the assessment of home cannabis cultivation as a criminal activity should be reconsidered,” or something like that.)
  2. 12 line 434-5. “Conversely, this practice should not be considered as a criminal activity.” Same issue as above. Also, ‘conversely’ seems odd in this context and I would drop the ‘as’.

Language issues

  1. 6 table 1. “Daily number of intakes (continue)”. I suppose the word ‘continue’ is here intended to inform the reader that this variable is a continuous one, and if so I would replace it with the latter.
  2. 9 line 291. “inprevious”. Typo.
  3. 11 line 362 ‘confirm’. I would recommend softening this statement by substituting ‘support’.
  4. 11 line 395. “In jurisdictions were home”. Typo ‘were’ (where).
Comments on the Quality of English Language

generally fine, but see above for a few small issues

Author Response

Comment 1 : 11 line 375-6. “home cultivation for personal use should not be considered as a criminal activity”. As much as I agree with this statement, it seems odd to present it basically out of nowhere. Why should it not be thus considered? Please explain or soften the categorical policy recommendation somewhat (e.g., “In context of the findings from these surveys, the authors believe that the assessment of home cannabis cultivation as a criminal activity should be reconsidered,” or something like that.)

Response 1 : Thank you for this comment. We agree that this statement could be better argued and more nuanced. Accordingly, we adapted it as follows (page 11 lines 379-385): “On the basis of these results, understanding this practice solely as a criminal activity would fail to consider its possible benefits and the motivations of home cultivators. We consider that the criminalization of home cultivation of cannabis for personal use and associated legal sanctions (e.g., in France, up to 7,500,000 euros fines and 20 years in prison) are inappropriate and disproportionate, precisely because of the fact that many home cultivators seek to distance themselves from the illegal market and its harmful aspects.

Comment 2 : 12 line 434-5. “Conversely, this practice should not be considered as a criminal activity.” Same issue as above. Also, ‘conversely’ seems odd in this context and I would drop the ‘as’.

Response 2 : Similarly, we developed and nuanced this statement as follows (page 13 lines 463-465): Consequently, considering this practice to be solely a criminal activity fails to understand its possible benefits and the motivations of home cultivators.”.

Comment 3 : 6 table 1. “Daily number of intakes (continue)”. I suppose the word ‘continue’ is here intended to inform the reader that this variable is a continuous one, and if so I would replace it with the latter.

Response 3 : Thank you for pointing out this error. We have corrected this error, replacing continue with “Continuous” (in two places) in Table 1 on page 6.

Comment 4 : 9 line 291. “inprevious”. Typo.

Response 4 : Thank you for pointing out this error. We corrected this: “in previous” (page 10 line 312).

Comment 5 : 11 line 362 ‘confirm’. I would recommend softening this statement by substituting ‘support’.

Response 5 : Thank you for this suggestion. We changed the term “confirm” to “support” (page 11 line 368).

Comment 6 : 11 line 395. “In jurisdictions were home”. Typo ‘were’ (where).

Response 6 : Thank you for pointing out this error. We corrected it to “where” (page 12 line 404).

Reviewer 3 Report

Comments and Suggestions for Authors

Line 48 write the all abbreviation THC

Rephrase please line 51

Line 59 no need to capitalize the , also in that line every time its written and.. and…

Line 79 the abbreviation THC was already apperead

Repharse line 111 please the meaning is not clear

In the introduction, the authors did not clearly articulate the research problem that motivated their study. Instead, they merely presented a series of general background information without establishing a coherent line of reasoning that justifies the necessity of their research.

Also the home clutivitaion concept must be defined

Materials and Methods

Since the study relies on secondary data, it is important to clarify whether the primary data sources have already been published. Please specify the origin and availability of these data sets.

Results

In my opinion, the results should be presented as a unified section. Presenting them separately as "Cannavid 1" and "Cannavid 2" gives the impression that they are two distinct studies, which may create confusion for the reader.

The results described between lines 230 and 248 lack clarity regarding which table they refer to. Please indicate the specific table that corresponds to these findings.

Regarding "Cannavid 2", only descriptive results are presented. It would be useful to explain why no further analysis (e.g., inferential statistics or comparisons) was conducted for this part of the study.

In the Discussion section, the authors are expected to acknowledge and clearly state the limitations of their study.

Funding

There appears to be a contradiction in the funding statement. The authors declare that the research received no external funding, yet they list multiple sources of support, including doctoral grants and institutional funding. Please clarify this discrepancy and clearly distinguish between personal (doctoral) funding and research project funding, if applicable

Comments on the Quality of English Language

some sentences needs to be rephrased for a smooth understanding

Author Response

Comment 1 : Line 48 write the all abbreviation THC

Response 1 : Thank you for pointing out this error. We now put “Δ-9-tétrahydrocannabinol before the abbreviation “THC” the first time it appears (page 2 line 49).

Comment 2 : Rephrase please line 51

Response 2 : We have done this. The new wording is as follows: “While the majority of cannabis produced and sold in Europe comes from medium-to-large production sites, the number of people growing small amounts of cannabis independently, with no connection to the illegal market, is increasing.” (page 2 lines 52-54).

Comment 3 : Line 59 no need to capitalize the , also in that line every time its written and.. and…

Response 3 : Thank you; we removed the capital letter from the word “the” (page 2 line 62). Concerning the words “and...”, this is because some of the quoted references mention two or more countries; we have removed the “and...”, now separating country names with a comma, and duplicated the references (page 2 lines 62-64).

Comment 4 : Line 79 the abbreviation THC was already apperead

Response 4 : Thank you; we removed the full word “Δ-9-tétrahydrocannabinol” and left the abbreviation (page 2 line 91).

Comment 5 : Repharse line 111 please the meaning is not clear

Response 5 : This has been rephrased to: “Despite all the above-mentioned data, there is a lack of data on the characteristics and motivations of home cultivators in France.” (page 3 lines 124-125).

Comment 6 : In the introduction, the authors did not clearly articulate the research problem that motivated their study. Instead, they merely presented a series of general background information without establishing a coherent line of reasoning that justifies the necessity of their research.

Response 6 : Thank you for this comment. The research problem is stated at the end of the introduction (page 3, lines 119–125). We agree that the initial structure of the introduction section did not provide a clear enough path to arrive at the research problem. The general background information was included to situate the study in both international and national contexts, which we feel is necessary to understand the rationale behind our research. Consequently, to improve readability of the introduction section and to make the objective clearer, we have added sub-sections and headings as follows: “Global cannabis market and home cultivation” (line 35), “Drug policy and cannabis use in France” (line 90), “Study objectives” (line 123).

Comment 7 : Also the home clutivitaion concept must be defined

Response 7 : Thank you for this suggestion. We added the following sentence (page 2 lines 54-55): “In this study, we refer to the practice of small-scale cultivation of cannabis primarily for personal use as ‘home cultivation’ ”.

Comment 8 : Since the study relies on secondary data, it is important to clarify whether the primary data sources have already been published. Please specify the origin and availability of these data sets.

Response 8 : We confirm that the data used in the present study were collected through two separate surveys (CANNAVID 1 and CANNAVID 2), whose initial protocols and main findings were presented in a previous publication, as referenced in the Methods section (page 3, lines 132–138) and cited in reference 26. With regard to data availability, this is addressed in the “Data availability statement” at the end of the manuscript (page 13, lines 485–486), which states that “The data presented in this study are available on request from the authors. The data are not publicly available due to privacy and ethical restrictions.”.

Comment 9 : In my opinion, the results should be presented as a unified section. Presenting them separately as "Cannavid 1" and "Cannavid 2" gives the impression that they are two distinct studies, which may create confusion for the reader.

Response 9 : "Cannavid 1" and "Cannavid 2" are indeed two separate studies with two different datasets which we analyzed separately. Some of the questions of interest in this study were only present in the CANNAVID 2 survey questionnaire. These questions were added to this questionnaire to provide complementary data.  Although mentioned in the abstract section (page 1 lines 17-19), we forgot to do so in the methods section. We added the following sentence in the methods section (page 5 lines 197-200): “We analyzed the two datasets (i.e. CANNAVID 1 and CANNAVID 2) separately. They could not be merged as we were not able to identify whether some of the participants had taken part in both surveys. In addition, some of the questions explored in the present study were specific to the CANNAVID 2 survey questionnaire.”.

Comment 10 : The results described between lines 230 and 248 lack clarity regarding which table they refer to. Please indicate the specific table that corresponds to these findings.

Response 10 : Thank you for this suggestion. We added this information at the beginning of the mentioned paragraph (page 7 line 251), as follows: “As shown in Table 1, participants reporting home cultivation as their main supply source mostly identified as men (86.4%) and […]”.

Comment 11 : Regarding "Cannavid 2", only descriptive results are presented. It would be useful to explain why no further analysis (e.g., inferential statistics or comparisons) was conducted for this part of the study.

Response 11 : We agree that it would have been interesting to merge the two datasets from the two seperate studies Cannavid 1 and Cannavid 2 in order to conduct larger inferential statistics and comparisons between both. However, we decided not to do so as we we were not able to identify whether some of the participants had taken part in both surveys. Therefore, we used only the larger of the two samples (Cannavid 1) to conduct the inferential statistics. In addition, we used the Cannavid 2 sample, mainly because the questionnaire contained additional questions (i.e., questions not in the Cannavid 1 survey) about participants’ home cultivation patterns and motives. The description of these results complemented the analysis of Cannavid 1’s dataset. We added a sentence to explain these points (also in response to a previous comment), as follows (page 5 lines 197-200): “We analyzed the two databases (i.e. CANNAVID 1 and CANNAVID 2) separately. They could not be merged as we were not able to identify whether some of the participants had taken part in both surveys. In addition, some of the questions explored in the present study were specific to the CANNAVID 2 survey questionnaire.”.

Comment 12 : In the Discussion section, the authors are expected to acknowledge and clearly state the limitations of their study.

Response 12 : Thank you for this suggestion. We initially discussed the limitations of the study at the beginning of the discussion section. However, we now create a separate sub-section at the end of the discussion section, specifically stating the limitations (page 12 line 430-450): “Limitations. This study has limitations. First, because the recruitment process used specialized websites, and because we restricted inclusion to persons with daily use, our results cannot be generalized to all persons who cultivate cannabis at home in France. It is possible that some home cultivators may have limited access to the internet. It is also possible that non-daily users of cannabis (e.g., occasional users) grow cannabis at home. In addition, more than 6% (CANNAVID 1) and 8% (CANNAVID 2) of participants did not answer the ‘main source of supply’ question, suggesting that for some users, this is sensitive information. Accordingly, our results may underestimate the number of home cultivators in both surveys. Furthermore, we cannot exclude the possibility of misclassification of participants who grew their own cannabis but not enough for it to be considered their main source of supply.

Second, this study provides only a limited understanding of home cultivation practices. Because the study design was cross-sectional, we were not able to identify the direction of the association between participants’ characteristics and the reported main supply source. In addition, since describing home cultivation practices was not the initial objective of the two CANNAVID surveys, only a very small number of questions about cultivation patterns and associated perceptions were present in both surveys’ questionnaires. Further studies could provide a deeper understanding of both the temporal development of home cultivation in the individual consumption trajectory, and the perceptions of benefits and risks associated with this practice.”.

Comment 13 : There appears to be a contradiction in the funding statement. The authors declare that the research received no external funding, yet they list multiple sources of support, including doctoral grants and institutional funding. Please clarify this discrepancy and clearly distinguish between personal (doctoral) funding and research project funding, if applicable

Response 13 : Thank you for this comment. Yes, we confused personal funding with study funding. We have moved the paragraph about doctoral grants that M.B. received from institutions to the acknowledgments section (page 13-14 lines 490-495).

Round 2

Reviewer 1 Report

Comments and Suggestions for Authors

The authors adequately addressed all my issues. Good paper. 

Reviewer 3 Report

Comments and Suggestions for Authors

Thank you for addressing all the points discussed in round 1. In this revised version, I only recommend removing line 374 ('Implication for harm reduction and drug policy'). Otherwise, I have no further comments.